# Reviews and syntheses: Hidden Forests, the role of vegetated coastal habitats on the ocean carbon budget

Carlos M. Duarte

*King Abdullah University of Science and Technology (KAUST), Red Sea Research Center (RSRC), Thuwal, 23955-6900, Saudi Arabia*

*Correspondence to*: Carlos M. Duarte (carlos.duarte@kaust.edu.sa)

**Abstract.** Vegetated coastal habitats, including seagrass and macroalgal beds, mangrove forests and salt-marshes, form highly productive ecosystems, but their contribution to the global carbon budget remains overlooked and these forests remain "*hidden*" in representations of the global carbon budget. Despite being confined to a narrow belt around the shoreline of the world's oceans, where they cover less than 7 million km$^2$, vegetated coastal habitats support about 1% to 10% of the global marine net primary production, and generate a large organic carbon surplus, of about 40 % of their NPP, which is either buried in sediments within these habitats or exported away. Large, 10-fold uncertainties in the area covered by vegetated coastal habitats, along with variability about carbon flux estimates, result in a 10-fold bracket around the estimates of their contribution to organic carbon sequestration in sediments and the deep sea from 73 Tg C year$^{-1}$ to 866 Tg C year$^{-1}$, representing between 3% and 1/3 of oceanic $CO_2$ uptake. Up to 1/2 of this carbon sequestration occurs in sink reservoirs (sediments or the deep sea) beyond these habitats. The organic carbon exported that does not reach depositional sites subsidizes the metabolism of heterotrophic organisms. In addition to a significant contribution to organic carbon production and sequestration, vegetated coastal habitats contribute as much to carbonate accumulation as coral reefs do. Whereas globally-relevant, the magnitude of global carbon fluxes supported by salt-marsh, mangrove, seagrass and macroalgal habitats is declining due to rapid habitat loss, contributing to loss of $CO_2$ sequestration, storage capacity and carbon subsidies. Incorporating the carbon fluxes vegetated coastal habitats support into depictions of the carbon budget of the global ocean and its perturbations will improve current representations of the carbon budget of the global ocean.

## 1 Introduction

Accounts of the role of primary producers in the global oceanic carbon cycle traditionally focus on the role of planktonic photosynthetic organisms and ignore, altogether, the potential contribution of marine vegetated coastal habitats (e.g. Falkowski et al. 2000, Fig. 6.1 in Ciais et al. 2013). The tenacity in ignoring the contribution of marine macrophytes is surprising, as not only was a significant role for marine macrophytes in the global oceanic carbon cycle highlighted already in 1981 (Smith 1981), but estimates of their important role as globally-significant carbon sinks developed a decade ago (Duarte et al. 2005) led to the development of a promising new strategy for climate change mitigation (Nature Editorial

2016), termed *Blue Carbon*, based on the conservation and restoration of these habitats (Nelleman et al. 2009, McLeod et al. 2011, Duarte et al. 2013a). Moreover, the focus on *Blue Carbon* has also driven attention to other aspects of the contribution of marine vegetated coastal habitats to the oceanic carbon budget beyond carbon burial in sediments, including export of organic carbon from the coastal to the open ocean (Dittmar et al. 2006, Barrón et al. 2014, 2015, Krause-Jensen and Duarte 2016).

Current neglect of the role of marine vegetated coastal habitats in the global carbon budget is largely derived from the flawed rationale that since these habitats are restricted to a narrow belt around the shorelines, they cannot possibly have a significant global role when compared to the vast spans of open oceanic waters dominated by phytoplankton, where benthic macrophytes cannot thrive. In addition, incorporating marine vegetated coastal habitats into the global carbon budget is made complicated by difficulties in assigning specific sources to the organic carbon burial in their soils, which is often partially allochthonous (e.g. Kennedy et al. 2010). Further, marine vegetated coastal habitats lack the charisma of other coastal ecosystems, such as coral reefs, and have not received much interest by the general public nor, possibly as a consequence, much research funding to assess their global role (Duarte et al. 2009), a tendency that the current focus on *Blue Carbon* is helping to revert. Whereas the focus on *Blue Carbon* has provided a major impetus to assess the global relevance of marine vegetated coastal habitats in the global carbon budget, these efforts have only addressed the contributions of these habitats to organic carbon burial in sediments, and have addressed other significant contributions of these habitats to the carbon budget of the global ocean. Hence, vegetated coastal habitats represent *hidden forests*, as they form ecosystems supporting some of the tallest plants in the biosphere (e.g. up to 45 m long kelps) with similar functions in carbon cycling as forests have, but that are not yet being recognized, despite abundant supporting evidence, as relevant components of the global carbon cycle.

Here I provide an overview of the extent, biomass and production of vegetated coastal habitats and the evidence for their role in the global carbon cycle and discuss how integrating their role in the context of the global ocean leads to reconsider some of the elements of the *status quo* of the global ocean carbon budget (e.g. as represented in Fig. 6.1 in Ciais et al. 2013). I then discuss how changes to marine vegetated coastal habitats derived from local impacts and direct human intervention but also from the consequences of climate change would affect the contribution of vegetated coastal habitats to carbon budgets regionally and globally, and identify future research challenges.

**2 Global extent and production of vegetated coastal habitats**

Vegetated coastal habitats occur along the coasts of all continents, but their nature varies depending on latitude and substrate characteristics. Where the substrate consist of soft sediments, muddy or sandy, salt-marshes and mangroves typically occupy the intertidal zone, with mangroves dominating in the tropics and salt-marshes in the temperate zone, while seagrass occupy the subtidal, and sometimes the lower intertidal zone, down to the depth receiving about 1 % of the light incident in the surface (Duarte 1991, Duarte et al. 2007). Green algae may grow within seagrass meadows, with calcifying algae (e.g. *Udotea sp.*, *Padina* sp., *Halimeda* sp.) and *Caulerpales* dominating in the tropics and subtropics, and *Ulvales* in the temperate zone. Macroalgae dominate rocky shores, from the intertidal zone down to depths receiving about 0.01 to 0.5 % of the light incident in the surface, depending on growth form (Gattuso et al. 2006). Macroalgal habitats are typically dominated by brown algae, including kelp communities in temperate, subpolar and polar latitudes, by *Sargassum* and *Turbinaria* in the subtropical and tropical zone, and dominated by *Cystoseira* in warmer temperate waters. Intertidal communities are dominated by *Fucus* and *Ascophyllum* from temperate to arctic latitudes. Foliose and filamentous macroalgae often develop high biomasses in nutrient-rich, estuarine enviroments (Valiela 2015), developing massive blooms, known as green tides, in hypereutrohic Chinese coastal areas (e.g. Ye et al. 2011). Mangroves develop forests that range from dwarf, 2 m tall trees at the poleward edge of their distributional limits and in arid and karstic areas lacking riverine inputs, to very large trees, exceeding 30 m in height in the wet tropics (Quisthoudt et al. 2012). Kelps also develop submarine forests with fronds up to 45 m long, while the landscapes formed by salt-marshes and seagrasses correspond more to those characteristic of dense wet meadows on land, with leaf area index exceeding 8 $m^2$ of leaf per $m^2$ of seafloor covered (Bay 1984).

The global area occupied by coastal vegetated habitats can be estimated using top down or bottom-up approaches. The former constrain the global extent by imposing ceilings derived from limiting factors, such as light or substrate availability. Bottom-up approaches attempt to derive a canonical estimate of their global areal extent by adding up the documented area covered in different regions. Unfortunately, such canonical estimates are precluded, for most coastal vegetated habitats, by the fact that only a fraction of them have been mapped. Mangrove forests are the only habitat for which a bottom-up estimate of global extent that is accurate and resolved at the regional level is available. A quasi-canonical estimate of the global area occupied (in year 2,000) of 0.137 $10^6$ $km^2$ was produced based on a detailed inspection of remote sensing images (Giri et al. 2011). Surprisingly, there is no validated estimate, to the best of my knowledge, for the global area of salt-marshes, despite these can also be extracted from remote sensing products. The only estimate available derived, four decades ago, assess the global area of salt-marshes at 0.38 $10^6$ $km^2$ (Table 1), with an uncertainty of about 50 % (Woodwell et al. 1973). However, the salt-marsh area has only been documented for Canada, Europe, the USA and South Africa, adding only to 0.022 $10^6$ $km^2$ (Chmura et al. 2003), representing < 10% of the global area estimate, which accuracy remains highly uncertain. Likewise, there is a large uncertainty as to the area occupied by seagrass and macroalgae, with estimates ranging between 2 $10^6$ $km^2$ and 6.8 $10^6$ $km^2$ (Table 1). The minimum area of seagrass, based on the total documented area, is much lower, at 0.15 $10^6$ $km^2$ (Green and Short, 2003), with an estimate of the likely global seagrass

extent of 600,000 km$^2$ , which assumes that only ¼ of the extant global seagrass area has been documented (Duarte and Chiscano, 1999,  Table 1).  Gattuso et al. (2006) calculated the potential coastal area marine macrophytes may occupy on the basis of the assessment of light requirements for marine macrophytes and light penetration around the coastal ocean. This procedure resulted in an estimate of the coastal area receiving sufficient solar irradiance at the sea floor to support seagrasses

of 5.19 10$^6$  km$^2$ (Gattuso et al. 2006). This surface area is 35 times larger than the documented seagrass extension of 0.15 10$^6$  km$^2$ (Green and Short, 2003) and about 9 times larger than the estimated likely area covered by seagrasses, estimated at 0.6 10$^6$  km$^2$ (Table 1). Gattuso et al. (2006) also calculated the potential global extent of macroalgal habitats at 5.71 10$^6$ km$^2$ in the non-polar and Arctic regions, respectively. This is about 1 10$^6$ km$^2$ below the maximum area estimated by Charpy-Roubad and Sournia (1990), although this difference may be accounted for the area covered in polar regions, which

may be substantial (Krause-Jensen and Duarte 2014). Gattuso et al. (2006) estimate of the potential area covered by macroalgae exceeds their estimates of that occupied by seagrass, a consequence of the lower minimum light requirements of macroalgae compared to seagrass, which have to support considerable non-photosynthetic (root and rhizome) biomass (Duarte et al. 1998).

The great uncertainty in the area occupied by marine vegetated coastal habitats is compounded with the fact that this is a

dynamic property, as vegetated coastal habitats are experiencing significant losses derived from anthropogenic impacts (Duarte et al. 2013a). The area occupied by seagrass, mangroves and salt-marshes has declined greatly due to human occupation of the coastal zone, land reclamation, deforestation and eutrophication, resulting in global loss rates of about 1 % year$^{-1}$ for angiosperm-dominated ecosystems (0.7 to 3 % year$^{-1}$, depending on ecosystems, Duarte et al. 2009; Duarte et al. 2013a), twice as high as those reported for tropical forests (Duarte et al. 2009).  For instance, whereas the area occupied by

seagrass is likely to be 4 times larger than that mapped to-date, consideration of seagrass losses during the 20$^{th}$ century (Waycott et al. 2009) suggests that the more likely global area occupied by seagrass is now of only 0.35 10$^6$ km$^2$ (Table 1).

Early estimates of the global net primary production (NPP) of marine macrophytes assessed this to be at least 1 Pg C year$^{-1}$ (Whitaker and Likens 1973, de Vooys 1979, Smith 1981), within the broad range of current estimates of the net community production, NCP, of marine macrophytes (0.18 to 4.84 Pg C year$^{-1}$, Table 2), although the most likely value is 1.9 Pg C year$^{-1}$

$^{1}$, dominated by macroalgae (Table 2).  Recently, Krause-Jensen and Duarte (2016) propagated uncertainties in the areal extent and primary production of macroalgae to derive an estimate of NPP for macroalgae at 1.52 Pg C year$^{-1}$, with the 25 and 75 percentiles of this estimate at 1.02 and 1.96 Pg C year$^{-1}$. However, a similar exercise has not yet been attempted for other vegetated habitat types.  Hence, the total net community production of marine vegetated habitats spans a broad 10-fold range from a minimum of 0.4 to 5.4 Pg C year$^{-1}$ (Table 2), due to combinations of uncertainties in the areal extent, the

dominant source of uncertainty, and the average net primary production per unit area.  Their net primary production, however, represents between < 1 % to about 10% of marine net primary production globally (Duarte and Cebrián 1996).

**3 The fate of the production of vegetated coastal habitats**

The role of vegetated coastal habitats on the global carbon budget is not, however, reflected in their NPP, as the fraction of NPP that is recycled within the ecosystem, through consumption, decomposition and, ultimately, respiratory processes, supports no net carbon flux. Hence, the focus should not be on the NPP supported by vegetated coastal habitats, but on its fate (Duarte and Cebrián 1996). The net primary production of vegetated coastal habitats meets four possible fates, it may be

(1) consumed by herbivores and detritivores, helping support the biomass and production of coastal food webs, (2) remineralised through respiration or decomposition by microorganisms and metazoans, (3) buried in sediments, or (4) exported away from the vegetated coastal habitat (Duarte and Cebrián 1996). Based on available estimates, Duarte and Cebrián (1996) concluded that marine macrophytes export or bury about 40 %, of their NPP, on average, ranging from average values of 35.3 % for marsh plants to 43.9 % for macroalgae (Table 2).

Vegetated coastal habitats are, therefore, strongly autotrophic ecosystems, as they produce organic carbon far in excess of local requirements (Duarte and Cebrián 1996, Duarte et al. 2010; Table 2). Thus, they act as strong sinks for atmospheric $CO_2$, as reflected in $pCO_2$ values typically sub-saturated relative to atmospheric equilibrium above submerged canopies (Smith 1981, Gazeau et al. 2005), driving a net uptake of atmospheric $CO_2$. In contrast, other coastal marine habitats, such as coral reefs (Gattuso et al. 1998), and estuarine environments typically act as sources of $CO_2$ to the atmosphere (Gattuso et

al. 1998, Frakignoulle et al. 1998, Borges 2005).

A fraction of the excess carbon produced by vegetated coastal habitats accumulates in their sediments. Indeed, salt-marshes, mangroves and seagrass meadows have been shown to support organic carbon stocks (Donato et al. 2011, Fourqurean et al. 2014) and burial rates (Duarte et al. 2005, 2013, McLeod et al. 2013) in the underlying sediments comparable or exceeding those supported by forests on land (Table 2). As a consequence, angiosperm-dominated coastal ecosystems have been

estimated to be responsible for 50% of the organic carbon burial, estimated at about 110 to 130 Tg C year$^{-1}$, in marine sediments, despite occupying only 0.2 % of the ocean area (Duarte et al. 2005). This estimate need be increased with a small contribution of about 6 to 10 Tg C year$^{-1}$ of carbon from macroalgae growing in soft sediments (Duarte and Cebrián 1996, Krause-Jensen and Duarte 2016). The estimate of the global burial of organic carbon in vegetated coastal habitats involves considerable uncertainties, compounding the large uncertainties in their global extent and NPP, discussed above, so the

estimates range 10 fold, from 0.044 to 0.404 Pg C year$^{-1}$ (Table 2).

Vegetated coastal habitats export, as terrestrial forest do, a significant fraction of their production. Organic carbon burial represents a modest, about 18 %, fraction of the net community production (NCP = burial + export in Table 2) of vegetated coastal habitats, dominated (55 % of total NCP) by export of marine macroalgae (Table 2). Hence, most (about 82 %) of the NCP of vegetated coastal habitats is exported, either as particulate or dissolved organic carbon (POC and DOC,

respectively). Tracking the fate of the export production of vegetated coastal habitats is, however, far more challenging than evaluating the carbon buried within their sediments. Carbon of coastal macrophytes can be tracked using a combination of stable isotope signatures, for seagrass and macroalgae, which are typically enriched in $^{13}$C relative to other primary

producers (Hemminga and Mateo 1996), specific organic markers, such as lipids, sterols and carotenoids, used mostly for macroalgae (Hardison et al 2013, Chikaraishi 2014), and, in principle, DNA barcoding approaches (Lucas et al. 2012, Nguyen et al. 2015), which may provide an unprecedented taxonomic resolution on the source of organic carbon, although these have not been tested to this end as yet.

A variable fraction of the exported material is deposited in the shores as beach-cast litter, with an important role in supporting terrestrial coastal food webs (Ochieng and Erftemeijer 1999, Ince et al. 2007, Mellbrand et al. 2011) and shoreline protection (Simeone and De Falco 2012, Boudouresque et al. 2015). Beach-cast deposits can reach phenomenal biomasses (Barreiro et al. 2011), such as up to 500 kg of dry wt m$^{-1}$ of shoreline of *Posidonia oceanica* litter washed on the shores of Tabarca Island, Spain (Mateo et al. 2003). Beach-cast material supports high metabolic rates (Coupland et al. 2007)

and represents a significant subsidy to terrestrial food webs (e.g. Ochieng and Erftemeijer 1999, Ince et al. 2007, Mellbrand et al. 2011), particularly in arid shores (e.g. Pollis and Hurd 1996), but the paucity of estimates on fluxes precluded any assessment of the fraction of export material that ends up washed on shores globally. A study in a Kenyan lagoon estimated that 19% of seagrass NPP were supplied as beach cast litter (Ochieng and Erftemeijer 1999). In addition, some of the beach-cast material is entrained again in the sea during storms or extreme tides, so it may be only temporarily deposited on shore.

Much of the carbon exported from vegetated coastal habitats is released as dissolved organic carbon (DOC). Dittmar et al. (2006) reported a large export of DOC from Brazilian mangroves, and calculated that DOC export from mangrove ecosystems represents approximately 26.4 Tg C year$^{-1}$, accounting for 60% of the upper estimate of mangrove C export (Table 2), consistent with estimates by Bouillon et al. (2008). Barrón et al. (2014) compiled estimates of net DOC release by seagrasses to conclude that they release, on average, 16 to 30 Tg C year$^{-1}$ as DOC, and Krause-Jensen and Duarte (2016)

estimated the DOC released by macroalgae at 355 (range 194 to 486) Tg C year$^{-1}$. Unfortunately, there is no estimate of the DOC export by saltmarshes, but that released by mangroves, seagrass and macroalgae together accounts for about 30% of their total export flux (Table 2). Much of this DOC export maybe remineralized by bacteria, as DOC exported from the coastal ocean has been argued to subsidize excess respiration in oligotrophic, open-ocean communities (Barrón et al. 2015). Krause-Jensen and Duarte (2016) estimated that 1/3 of the DOC flux is exported, by vertical turbulent diffusive transport,

below the mixed layer, eventually reaching the deep sea (> 1,000 m), where some of it would be sequestered, as organic carbon entering the deep sea is removed from exchange from the atmosphere over centennial time-scales, thereby qualifying as sequestration independently of whether it is remineralised or not.

The bulk (about 70 %) of carbon export from vegetated carbon export is released as particulate organic carbon (POC). Some of the POC export is sequestered in depositional sites outside the vegetated coastal habitats, including sediments in the

continental shelf or the deep ocean. Krause-Jensen and Duarte (2016) reviewed available evidence on the presence of macroalgal carbon in shelf sediments and the deep sea to conclude that a total of about 14 Tg C year$^{-1}$ and 35 Tg C year$^{-1}$ of macroalgal POC are sequestered in continental shelf sediments outside macroalgal beds and the deep sea, respectively.

Hence, burial of macroalgal carbon beyond macroalgal habitats is at least four times greater than burial in macroalgal beds occurring in soft sediments. There is, unfortunately, no comparable estimate of the burial of exported mangrove, salt-marsh or seagrass POC beyond their habitats. However, reports of seagrass carbon in unvegetated sediments adjacent to seagrass meadows (e.g. Kennedy et al. 2010) and leaf litter on deep-sea sediments (e.g. Moore 1963, Wolff 1976) suggests that, as for macroalgae, seagrass carbon also reaches depositional sites outside seagrass meadows, representing a component of carbon burial yet to be quantified.

Krause-Jensen and Duarte (2016) estimated that 1/4 of the export flux of macroalgae is sequestered in unvegetated sediments or the deep sea. Assuming that the export flux of mangroves, salt-marshes and seagrass meadows meets a similar fate, would suggest that vegetated coastal habitats contribute to sequestration of about 29 Tg C year$^{-1}$ to 462 Tg C year$^{-1}$ beyond their habitats. Macroalgae, which had been largely neglected as components of marine carbon sequestration (Hill et al. 2015, Krause-Jensen and Duarte 2016), now emerge as main contributors to the role of vegetated coastal habitats in carbon sequestration (Krause-Jensen and Duarte 2016). Combining burial in blue carbon habitats with sequestration beyond them, indicates that vegetated coastal habitats sequester 73 Tg C year$^{-1}$ to 866 Tg C year$^{-1}$. Hence, vegetated coastal habitats would contribute between a minimum of 0.3% to a maximum of 1/3 of the biological $CO_2$ removal by marine biota estimated to represent about 2,000 Tg C year$^{-1}$, which had hitherto been attributed entirely to phytoplankton photosynthesis in depictions of the global carbon budget (Fig. 6.4 Ciais et al. 2013). Moreover, the carbon exported to the open ocean contributes to subsidize heterotrophic metabolism in open ocean communities, contributing to support the excess community respiration over production often encountered in the oligotrophic ocean (Duarte et al. 2013b, Barrón et al. 2015).

The estimates above all refer to organic carbon, the component of the ocean carbon budget that has been the focus of carbon assessments in the framework of climate change (Ciais et al. 2013). However, vegetated coastal habitats are also important sites for carbonate formation and dissolution, although information on the global fluxes involved have received even less attention than that of organic carbon fluxes. Calcareous algae, such as coralline and *Halimeda*, have been long recognized to be important contributors to carbonate formation, with estimates of net calcification by calcifying algae being in the order of 20 Tg C year$^{-1}$ for *Halimeda* bioherms (Milliman and Droxler 1996). The carbonate production in seagrass meadows was recently estimated at 20 to 75 Tg C year$^{-1}$ (Mazarrasa et al. 2015). There is no information of the carbonate deposition in mangrove or salt-marsh sediments, probably due to the believe that they are unlikely to accumulate carbonate. However, mangroves have also been reported to develop carbonate soils (e.g. Koch and Snedaker 1997) so even if small there must be some contribution from mangroves, and, likely, salt-marshes. Hence, carbonate accumulation in vegetated coastal sediments is likely to be, at least, comparable to that of coral reefs (> 40 to 95 Tg C year$^{-1}$ in vegetated coastal sediments vs. 84 Tg C year$^{-1}$ for coral reefs, Milliman and Droxler 1996).

As carbonate production in shallow waters results in the release of 0.63 mols of $CO_2$ per mol of $CaCO_3$ precipitated (Smith 2013), the accumulation of $CaCO_3$ in vegetated coastal sediments could be considered to offset carbon sequestration by 25 to

60 Tg C year$^{-1}$, thereby reducing organic carbon sequestration in vegetated coastal habitats. However, this simple interpretation considers carbonate and organic burial to be independent, which may be incorrect. In particular, organic matter tends to be closely associated with $CaCO_3$ particles, becoming less accessible to remineralization by microorganisms, resulting in significantly greater $C_{org}$ preservation in carbonate-rich sediments (Mayer 1994). Moreover, remineralization of

sediment organic matter increases $CO_2$ and may lead to carbonate dissolution, which would in turn lead to $CO_2$ removal (Smith 1981), so co-deposition of organic and inorganic carbon may buffer against $CO_2$ release of disturbed sedimentary deposits. Overall, our understanding of the carbonate budget of vegetated coastal habitats lags well behind that of organic carbon, with which it likely interacts rather than being just a parallel, independent process.

## 4 Future Trends and Research Needs

Resolving the uncertainties on the global area covered by salt-marsh, seagrass and macroalgal habitats and its regional distribution is an imperative, as these uncertainties remain the largest source of uncertainty as to their role in the global carbon cycle. The rise of interest in *Blue Carbon* strategies has led to an increase in the data available on organic carbon stocks and burial rates in vegetated coastal habitats, including efforts to improve the representation of vegetated coastal habitats outside North America, Europe and Australia, where the majority of the estimates come from. However, the fate of

the large export flux remains unaccounted for, with a first-order assessment available only for macroalgal carbon (Krause-Jensen and Duarte 2016), which are, however, responsible the largest export flux.

The large uncertainties as to the global extent of vegetated coastal habitats are compounded with its rapid change, as these habitats experience some of the steepest rates of any ecosystem, at loss rates of 0.7 to 3 % year$^{-1}$, depending on ecosystems (Duarte et al. 2008, Duarte et al. 2013a, Waycott et al. 2009), two to ten times greater than that of tropical forests. These

losses are largely attributable to local anthropogenic perturbations, such as mechanical destruction in converting them into aquaculture ponds, urban areas and other uses, eutrophication and other perturbations (Duarte et al. 2002, Waycott et al. 2009). However, climate change plays an increasingly larger role, leading to shifting biogeographical ranges, generally involving losses in the equator-ward ranges (e.g. Wernberg et al. 2010, Moy and Christie 2012, Tanaka et al. 2012, Voemann et al. 2013) and poleward migration at the poleward edge, which for macroalgae occurs at characteristic rates of about 30

km decade$^{-1}$ (Poloczanska et al. 2014). The prospect for poleward kelp expansions is particularly significant for the Arctic, which convoluted coastline would offer a large habitat for kelps in a rapidly warming Arctic (Krause-Jensen and Duarte 2014). In addition, macroalgal aquaculture has emerged as a globally-significant activity with a yield of 26.9 million tons (dry weight) in 2013, and growing at a rate of 7.9 ± 0.2 % year$^{-1}$ (data from www.fao.org/figis , accessed 21 November, 2015). This represents a production of about 10 Tg C year$^{-1}$, about 5 % of global seaweed production (Table 2). Whereas

POC export from macroalgal farms will likely be greatly reduced compared to wild stocks as macroalgae are harvested, macroalgal crops should export comparable DOC to wild stocks, along with some POC, thereby likely contributing to enhance the role of macroalgae in carbon export and sequestration.

The large changes in the area covered by vegetated coastal habitats, with at least 1/3 of the global cover already lost, together with their significant contribution to carbon cycling indicate that perturbations to vegetated coastal habitats should contribute to the components of green-house emissions termed "*land-use change*" sources, although this has not been accounted for. A third of the loss in the global biomass of marine macrophytes of about 1 Pg C (Smith 1981), one of the components of vegetated coastal habitats, would have contributed about 0.33 Pg C to accumulated emissions. However, the emissions derived from the erosion of the large carbon stocks under disturbed vegetated coastal habitats are potentially much greater, at about 0.12 Pg C yr$^{-1}$ (Pendelton et al. 2012). Assessments of the realized cumulative green-house gas emissions due to disturbance of vegetated coastal habitats and the risks of further emissions from future disturbance should be incorporated into accounts of realized perturbations to the global carbon budget and scenarios of possible future perturbations. Moreover, these assessments, still pending, are essential to evaluate the potential global benefits of restoration and conservation measures to recover and avoid the loss of these intense carbon sinks. Lastly, evidence of the major export of organic carbon from vegetated coastal habitats to the open ocean should prompt research, assisted by the availability of more and more powerful markers, to elucidate its role in the functioning of the open ocean and deep-sea ecosystems, a role that was considered significant already fifty years ago (cf. Krause-Jensen and Duarte 2016).

Whereas estimates of offshore export of "blue carbon" are now becoming available (Cebrián and Duarte 1986, Duarte et al. 2005, Dittmar et al. 2006, Barrón et al. 2014, 2015, Krause-Jensen and Duarte 2016), the exchange of carbon across the air-sea and land-ocean boundaries of vegetated coastal habitats remains poorly resolved. The strong autotrophic nature of macroalgal and seagrass habitats is further reflected in their role as strong sinks for atmospheric $CO_2$ (e.g. Gazeau et al. 2005, Unsworth et al. 2012, Antony et al. 2013, Tokoro et al. 2014, Ikawa and Oechel 2015). Seagrass, salt-marshes, macroalgae and mangroves all contribute significant loads of material to adjacent beaches, where they can accumulate large carbon stocks (e.g. Mateo et al. 2003, Simeone and de Falco 2012, Gomez et al 2013). However, they receive greater subsidies of plankton and land-derived "green carbon", which have been shown to comprise typically about 50% of the organic carbon stock in seagrass sediments (Kennedy et al. 2010). Hence, organic carbon input from offshore and land sources contribute to the large carbon burial capacity of vegetated coastal habitats while allowing them to export a significant fraction of their own production. Resolving the exchange of carbon between vegetated coastal habitats and adjacent marine, terrestrial and atmospheric components will help further constrain their local and global role in carbon budgets, as well as the consequences of losses or gains of these habitats on carbon flow.

**5 Conclusions**

Despite current uncertainties it is clear that future representations of the carbon budget of the coastal ocean should cease to ignore vegetated coastal habitats, or assume that this component is lumped within the term "*marine biota*" present in current representations (e.g. Ciais et al. 2013), which is not, as the associated fluxes and pools are those corresponding to marine

plankton.  The important role of vegetated coastal habitats in the carbon budget, contributing 1% to 10 % of oceanic net primary production (Smith 1981), 0.3 to 1/3 of the oceans' biological pump and from >0.6 % to 2/3's of carbon burial in sediments is now evident to scientists and policy makers and seems to be ignored only by global carbon budget modelers (e.g. Ciais et al. 2013), for which these habitats continue to be *hidden forests*.

5        Some years ago, a working group led by Jon J. Cole, Yves T. Prairie and I, synthesized available evidence to point at globally-significant organic carbon burial and $CO_2$ emissions from freshwater ecosystems (Cole et al. 2007). This effort led to these fluxes (200 and 1,000 Tg C year$^{-1}$, respectively) now being explicitly captured in the latest representation of the global carbon budget by the IPCC (Fig. 6.4, Ciais et al. 2013). The carbon fluxes dominated by the "hidden forests" of the coastal ocean are likely to be at least of a similar magnitude and should, therefore, be also captured in future representations

of the global carbon budget.  This will require an additional effort to improve the precision about current estimates. The uncertainty in the global area these habitats cover has not been narrowed down, for seagrass, macroalgae and salt-marshes, for several decades now, and the estimates of the global NPP contributed by these habitats and its fate have not been revisited since the estimates provided by Smith (1981) and Duarte and Cebrián (1996) several decades ago. As in the case of freshwater carbon emissions and burial, incorporating the carbon fluxes vegetated coastal habitats support into depictions of

the global carbon budget and its perturbations also requires that the research community addressing carbon fluxes in vegetated coastal habitats reach out to establish links to share knowledge on these fluxes with the working groups involved in assessing the global carbon budget.

*The author declares that he has no conflict of interest.*

**Acknowledgements**

This paper conveys my lecture in accepting the *Vladimir Ivanovich Vernadsky* Medal 2016 of the European Geophysical Union.  I thank the colleagues that nominated and supported me for this award and the many colleagues that have

collaborated in this research over the years, particularly Dorte Krause-Jensen, Nuria Marbá, Jack Middelburg, Jim Fourqurean, Paul Lavery, Miguel Angel Mateo, Peter Macreadie, Oscar Serrano, Pere Masqué, Inés Mazarrasa and Catherine Lovelock.

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

Table 1. Estimates of the global area covered by vegetated coastal habitats, indicating the level of confidence on the estimates and whether they represent lower or upper limit estimates.

| Habitat | Area ($10^6$ km$^2$) | Reference | Confidence | Notes |
|---|---|---|---|---|
| Mangroves | 0.137 | Giri et al. (2011) | High | 1 |
| Salt-marshes | 0.02 | Chmura et al. (2003) | Lower limit | 2 |
| | 0.38 | Woodwell et al. (1973) | Low | 3 |
| Seagrass | 0.15 | Green and Short (2003) | Lower limit | 4 |
| | 0.35 | Duarte et al. (2005) | Low | 5 |
| | 0.6 | Duarte and Chiscano | Upper limit | 6 |
| | 4.32 | Gattuso et al. (2006) | Upper limit | 7 |
| Macroalgae | 1.4 | Duarte et al. (2013) | Low | 8 |
| | 2 | Gattuso et al. (1998) | Upper limit | 9 |
| | 3.4 | Charpy-Roubad and Sournia (1990) | Low | 10 |
| | 5.71 | Gattuso et al. (2006) | Upper limit | 11 |
| | 6.8 | Charpy-Roubad and Sournia (1990) | Upper limit | 12 |

1. Global assessment of Landsat satellite images for year 2,000

2. Based on documented area in Canada, Europe, the US, and South Africa

3. Estimated based on the fraction of coastaline occupied by estuaries and assuming 20% of the area of estuaries to be salt marsh

4. Derived by combining the seagrass area documented regionally

5. Assumes that about half of the potential area has been lost

6. Assumed documented area to be 1/4 of total area

7. Gattuso et al. (2006) combined estimates of underwater light penetration, global bathymetry and the light requirements of segrass to estimate the potential area available for seagrass

8. Substracts the likely seagrass area from Duarte and Chiscano from the total macrophyte area in Gattuso et al. (1998)

9. Area of estuaries, algal beds and reefs from Table 1 in Whitaker and Likens (1973) used by Gattuso et al. (1998) to represent global macrophyte (seagrass + macroalgae) area

10. Charpy-Roubad and Sournia (1990) consider that only half of the potential area (6.8 $10^6$ km$^2$) is occupied.

11. Gattuso et al. (2006) combined estimates of underwater light penetration, global bathymetry and the light requirements of macroalgae to estimate the potential area available for macroalgae.

12. Estimated as the potential area available for macroalgae based on a literature review.

Table 2. Net primary production (NPP), carbon burial and export production of vegetated coastal habitats. Lower range of areal production values from Duarte and Chiscano (1999) and upper range of areal seagrass production calculated from gross community production in Gatuso et al. (1998), assuming community respiration (R) R=0.5·GPP from Duarte and Cebrián (1996). Upper value for areal mangrove and salt-marsh production calculated as the ratio between global NPP and global area in Duarte and Cebrián (1996). Range of global macroalgal production from Krause-Jensen and Duarte (2016). Percent NPP buried and exported for various habitats from Duarte and Cebrian (1996), and global burial and export ranges calculated by combining these percent values with the range of global NPP values.

| Habitat | NPP | | Burial | | Export | |
|---|---|---|---|---|---|---|
| | $g\,C\,m^{-2}\,y^{-1}$ | Range Pg C $y^{-1}$ | % NPP | Range Pg C $y^{-1}$ | % NPP | Range Pg C $y^{-1}$ |
| Seagrass | 394 - 449 | 0.06 - 1.94 | 15.9 | 0.01 - 0.308 | 24.3 | 0.014 - 0.471 |
| Macroalgae | 9 1- 522 | 0.127 - 2.9 | 0.4 | 0.0005 -0.012 | 43.5 | 0.055 - 1.26 |
| Salt-marsh | 438 -1100 | 0.17 - 0.42 | 16.7 | 0.028 - 0.070 | 18.6 | 0.031 - 0.078 |
| Mangroves | 394 - 1000 | 0.05 - 0.15 | 10.4 | 0.005 - 0.016 | 29.5 | 0.014 - 0.044 |
| Total | | 0.407 - 5.41 | | 0.044 - 0.404 | | 0.116 - 1.85 |

