# Peer review of "Reviews and syntheses: Hidden Forests, the role of vegetated coastal habitats on the ocean carbon budget"

_Biogeosciences, 2016_

## Referee Comment (RC1) · Anonymous Referee #1 · 9 Sep 2016

Review of Duarte – Hidden Forests

This manuscript on the role of vegetated coastal ecosystems in the ocean carbon budget is generally fine, but in my opinion does not really offer much novelty or new insights compared to a number of earlier syntheses on the same topic. The strongest point is the emphasis on the uncertainty in the area covered by these different types of ecosystems, and on the implications this has for their estimated global carbon fluxes – but these uncertainties are not consistently applied. I don't have a problem seeing this published but do feel the added contribution to existing literature is rather slim unless some new aspects are included. That being said, I do have a number of suggestions for improvement or to increase the consistency; I have listed these below.

The author correctly emphasizes the uncertainty in global areal estimates of vegetated coastal ecosystems, and that this 10-fold uncertainty implies and equally large uncertainty in e.g. global OC sequestration rates or production rates. However, I do not see this consistently emphasized in the latter data, and I suspect that the numbers that will be picked up from this work and cited later on are the maximum potential fluxes/rates – the way these are presented is somewhat biased then. To illustrate my point:

Page 1, Line 16: "representing up to 1/3 of the biological CO2 removal by marine biota". OK – but given the 10-fold range in areas, one could also write "representing as little as 3.5 % of the biological CO2 removal by marine biota" if we take the lower value of areal rates ? I'm obviously not advocating for the latter, but if the uncertainty brackets a 10-fold range, I don't feel it's fair to mention only the maximum values in abstract and conclusions, just to stress the potential importance of these ecosystems and to raise awareness. The same issue in th Conclusions, page 9 line 9-11: "contributing 10% of the oceanic NPP, 1/3 of the ocean's biological pump and >2/3 of carbon burial of sediments is now evident"

Page 4, line 19-20: NPP is ∼10% of marine net primary production globally. You refer here to Duarte & Cébrian (1996), further on to Smith (1981) for the same statement (page 9, line 10). Both are somewhat older publications, aren't there new data to revise this estimate (read: should this not be one of the objectives of this paper) and aren't those estimates based on a fixed and highly uncertain areal extent as well ? It is somewhat counterintuitive to stress the uncertainty in the role of these systems in the global (ocean) C budget due to the uncertain global areal cover, but to stick to a fixed contribution to marine NPP based on syntheses performed >20 years ago.

Page 7, line 5-10: "Hence, vegetated coastal habitats would contribute up to 1/3 of the biological CO2 removal by marine biota estimated to represent about 2000 Tg C y-1, which had hitherto been attributed entirely to phytoplankton photosynthesis (Ciais et al 2013). Several points/suggestions regarding this statement:

1/ up to 1/3rd of biological $CO_2$ removal by marine biota: again, this is stressing the upper limit, see comment above. One could take the opposite view and claim they contribute as little as 1/30th ? 2/ Unless I'm mistaken, the numbers by Ciais refer to net $CO_2$ uptake by the global ocean, it does not claim that this $CO_2$ drawdown is entirely due to phytoplankton production ? 2/ Vegetated coastal ecosystems such as mangroves and salt marshes (and subtidal seagrass beds to a certain extent) take up $CO_2$ from the atmosphere, not from the ocean water. Hence, their productivity would not directly lower $pCO_2$ in the ocean and will not lead to $CO_2$ uptake by the ocean, in that context comparing NPP data from all vegetated coastal ecosystems combined is difficult to compare directly with data on net ocean $CO_2$ uptake.

Page 9, line 10: coming back to Smith (1981): while it's good to acknowledge the early work, I doubt this should be used as the most recent / best estimate of the contribution of these ecosystems to NPP. There are many more datasets published in the meantime, and Smith (1981) used a fixed area of 200,000 $km^2$ and included only seagrasses and macroalgae. If the objective is to provide a state-of-the-art, use the best estimates available + include the uncertainty which is a key message elsewhere in the paper.

-Table2 should be clarified:

1/ "Lower range of production values from Duarte etc": specify whether this refers to the areal rates (first column) or the total NPP range (2nd column).

2/ "Upper value for mangrove and salt marsh production calculated as the ratio between global NPP and global area in Duarte & Cebrian": this does not make sense to me. The global NPP data in Duarte & Cébrian were calculated assuming a certain area for each of these ecosystems, taking those globally integrated NPP values and dividing them by - I assume – a range of (different) area estimates is not defendable. Or perhaps I misunderstand what was done to derive these numbers – explain in more detail.

3/ % buried and exported, data from Duarte & Cébrian (1996). Here too, can these estimates not be easily refined given the large amount of studies performed in the 20

years since this publication ?

Minor issues:

-be consistent in using km$^2$ and not Km$^2$

-Page 2 line6-9: Is an alternative reason not that fitting them into the global C budget is also made complicated by the fact that they are complex ecosystems at the land-ocean interface, and that flux measurements (e.g. OC burial rates in seagrass beds, to name but one) are somewhat complicated to assign to specific sources/origin, e.g. much of the OC burial in seagrass beds may be terrestrial or mangrove carbon. This paragraph is perhaps a little to pessimistic about the recognition they receive, given the strong impetus in studies on C cycling in vegetated coastal ecosystems during the past 15-20 years.

-page 8 line 16: poleword (not poelword)

-page 8 line 9: loss rates of 0.5 – 5 % year-1, this is a different range of loss rates than that cited on page 4 line 9. Use consistent numbers and references.

-page 8, last line: "Lastly, realization of the major export of organic matter [. . .]": what is meant by this?

-page 9, first line: "available" should be "availability"

---

## Referee Comment (RC2) · Anonymous Referee #2 · 25 Sep 2016

General comments:

This manuscript is a significant addition to the author's previous 4 reviews and syntheses on the role of vegetated shallow coastal ecosystems in carbon sinks (Nellemann et al., 2009; McLeod et al., 2011: Duarte et al., 2013a; Krause-Jensen and Duarte, 2016); in particular, a new inclusion of estimates of "off-site blue carbon sink", i.e., carbon export from coastal habitats and subsequent sequestration in the deep sea (Table 2).

The manuscript is generally well-written and easy to follow; however, I found numerous inconsistencies between the citation in the text and the reference list, which should be corrected.

[Figure]

The result was a surprising because the author estimated that, by including previously overlooked macroalgal contributions, the carbon export far exceeds carbon burial, which is previously thought to be the major mechanism for coastal carbon sequestration. Although, as the author mentioned, this estimate has a large 10-fold uncertainties and not well constrained at this stage, this manuscript would potentially revise our recognition on the role of vegetated coastal habitats on carbon sequestration.

In terms of carbon budget in vegetated coastal habitats, this synthesis still lacks quantification of two boundary conditions (terrestrial inputs and air-water carbon exchanges) and their effects on the carbon sequestration. However, my understanding is that we have large research gaps in this two components and can be mentioned in the Future Trends and Research Needs section, although several recent papers have been dealing with air-sea CO2 exchanges in such a vegetated shallow waters and other papers have shown the separated estimation of green (terrestrial allochthonous) and blue (autochthonous) carbon in the sediments.

Specific comments:

P17 Table2: I understand that the estimates of Export is new; however, are the estimates of NPP and Burial new? i.e, are these estimates overall the same as the previous your publications (Nellemann et al., 2009; McLeod et al., 2011: Duarte et al., 2013a; Krause-Jensen and Duarte, 2016) and Pendleton et al. (2012) or revised values? If revised, how much is the differences compared to the previous estimates?

Technical corrections:

P2L11, P4L8, P: Duarte et al. 2009 is missing in the Reference

P2L26: Duarte et al. 1991 is missing in the Reference

P2L29: Gattuso et al. 2006, not 2006 but 2005?

P3L19: Chmura et al, 2003 is missing in the Reference

[Figure]

P4L10: Waycott et al 2006 is missing in the Reference, 2009?

P4L12: Whitaker and Likens 1973 is missing in the Reference

P5L2: Duarte et al. 2011 is missing in the Reference

P5L8: Fourqurean et al. 2014 is missing in the Reference

P5L9, P8L10: Duarte et al. 2013, "a" or "b" here?

P5L32: Coupland et al. 2007 is missing in the Reference

P7L16: Mazarrasa et al. 2015 is missing in the Reference

P7L23: Smith 2013 is missing in the Reference

P7L28: May 1994 is missing in the Reference

P8L12: Waycott et al. 2011 is missing in the Reference, 2009?

P7L28: Duarte et al. 2002 is missing in the Reference

P9L30: Macreadie?

P10L18: not cited in the text

P11L22: not cited in the text

P12L4: not cited in the text

P13L13: not cited in the text

P14L8: not cited in the text

P15L4: not cited in the text

---

## Author Comment (AC1) · 20 Dec 2016

Dear Dr. Middelburg,

Below, please find my response to the comments raised by the reviewers to my manuscript entitled "Reviews and syntheses: Hidden Forests, the role of vegetated coastal habitats on the ocean carbon budget", along with a description of the changes made in the revised manuscript, which has been uploaded in the system.

I believe that, as an outcome of my efforts to address the reviewers' comments, the revised manuscript is much improved and, I hope, you will now find it suitable to be accepted for publication in Biogeoscience.

I have prepared the revised version with the changes described in the response below and I am eager to upload it as soon as you inviteme to do so.

Sincerely,

Carlos M. Duarte

Changes made in response to comments by Anonymous Referee 1

This manuscript on the role of vegetated coastal ecosystems in the ocean carbon budget is generally fine, but in my opinion does not really offer much novelty or new insights compared to a number of earlier syntheses on the same topic. The strongest point is the emphasis on the uncertainty in the area covered by these different types of ecosystems, and on the implications this has for their estimated global carbon fluxes – but these uncertainties are not consistently applied. I don't have a problem seeing this published but do feel the added contribution to existing literature is rather slim unless some new aspects are included. That being said, I do have a number of suggestions for improvement or to increase the consistency; I have listed these below.

Comment: Whereas there have indeed been a number of review articles on the topic of blue carbon recently, these address the magnitudes of stocks and CO2 burial in vegetated coastal habitats, but do not address the other components of their role in the ocean carbon cycle nor make or support the case that these ecosystems merit explicit consideration in future depictions of the global ocean carbon cycle. As is, these depictions still ignore these important habitats and refer to phytoplankton as the sole relevant authotchtonous source of organic carbon in the marine environment. Hence, the need to make this plea explicit.

Moreover, the series of reviews or global estimates of Blue Carbon all have carried over estimates of the area occupied by these ecosystems, which is central to the up-scaling and the global figures delivered, without addressing the uncertainties around these values, which have been propagated along citation chains, unchallenged, for almost

three decades.

In any case the reviewer's suggestions have resulted in a much stronger and focused paper, which now provides, I believe, a stronger addition to the existing literature.

Action: I have now explicitly indicated, in the introduction, that "Whereas the focus on Blue Carbon has provided an additional basis to assess the global relevance of marine vegetated coastal habitats in the global carbon budget, these efforts have only addressed the contributions of these habitats to organic carbon burial in sediments, and have addressed other significant contributions of these habitats to the carbon budget of the global ocean.".

The author correctly emphasizes the uncertainty in global areal estimates of vegetated coastal ecosystems, and that this 10-fold uncertainty implies and equally large uncertainty in e.g. global OC sequestration rates or production rates. However, I do not see this consistently emphasized in the latter data, and I suspect that the numbers that will be picked up from this work and cited later on are the maximum potential fluxes/rates – the way these are presented is somewhat biased then. To illustrate my point: Page 1, Line 16: "representing up to 1/3 of the biological $CO_2$ removal by marine biota". OK – but given the 10-fold range in areas, one could also write "representing as little as 3.5 % of the biological $CO_2$ removal by marine biota" if we take the lower value of areal rates ? I'm obviously not advocating for the latter, but if the uncertainty brackets a 10-fold range, I don't feel it's fair to mention only the maximum values in abstract and conclusions, just to stress the potential importance of these ecosystems and to raise awareness. The same issue in the Conclusions, page 9 line 9-11: "contributing 10% of the oceanic NPP, 1/3 of the ocean's biological pump and >2/3 of carbon burial of sediments is now evident"

Action: I agree, and I have, therefore, carried over the uncertainty into the various estimates. The text now reads: "Large, 10-fold uncertainties in the area covered by vegetated coastal habitats, along with variability about carbon flux estimates, result

in a 10-fold bracket around the estimates of their contribution to organic carbon sequestration in sediments and the deep sea from 73 Tg C year-1 to 866 Tg C year-1, representing between 3% and 1/3 of oceanic CO2 uptake. ", "Hence, the total net community production of marine vegetated habitats spans a broad 10-fold range from a minimum of 0.4 to 5.4 Pg C year-1 (Table 2), due to combinations of uncertainties in the areal extent, the dominant source of uncertainty, and the average net primary production per unit area. Their net primary production, however, represents between < 1 % to about 10% of marine net primary production globally (Duarte and Cebrián 1996).", and "The important role of vegetated coastal habitats in the carbon budget, contributing 1% to 10 % of oceanic net primary production (Smith 1981), 0.3 to 1/3 of the oceans' biological pump and from >0.6 % to 2/3's of carbon burial in sediments is now evident to scientists and policy makers and seems to be ignored only by global carbon budget modelers (e.g. Ciais et al. 2013), for which these habitats continue to be hidden forests. Some years ago, a working group led by Jon J. Cole, Yves T. Prairie and I, synthesized available evidence to point at globally-significant organic carbon burial and CO2 emissions from freshwater ecosystems (Cole et al. 2007). This effort led to these fluxes (200 and 1,000 Tg C year-1, respectively) now being explicitly captured in the latest representation of the global carbon budget by the IPCC (Fig. 6.4, Ciais et al. 2013). The carbon fluxes dominated by the "hidden forests" of the coastal ocean are likely to be at least of a similar magnitude and should, therefore, be also captured in future representations of the global carbon budget. This will require an additional effort to narrow down the large uncertainty in the global area they cover.".

Page 4, line 19-20: NPP is _10% of marine net primary production globally. You refer here to Duarte & Cébrian (1996), further on to Smith (1981) for the same statement (page 9, line 10). Both are somewhat older publications, aren't there new data to revise this estimate (read: should this not be one of the objectives of this paper) and aren't those estimates based on a fixed and highly uncertain areal extent as well ? It is somewhat counterintuitive to stress the uncertainty in the role of these systems in the global (ocean) C budget due to the uncertain global areal cover, but to stick to a fixed

contribution to marine NPP based on syntheses performed >20 years ago.

Action: Unfortunately, efforts to improve our estimate of global seagrass and macroalgal NPP are still affected by the uncertainties in the area they cover, so even though the references used are "old", we can only offer brackets encompassing those estimates at present. I now acknoedge this uncertainty throughout. The text has been revised to acknowledge these uncertainties. The text now reads: "vegetated coastal habitats support about 1% to 10% of the global marine net primary production,.." (and see above), "Their net primary production, however, represents between < 1 % to about 10% of marine net primary production globally (Duarte and Cebrián 1996)."; "The important role of vegetated coastal habitats in the carbon budget, contributing 1% to 10 % of oceanic net primary production (Smith 1981), 0.3 to 1/3 of the oceans' biological pump and from >0.6 % to 2/3's of carbon burial in sediments is now evident to scientists and policy makers and seems to be ignored only by global carbon budget modelers (e.g. Ciais et al. 2013), for which these habitats continue to be hidden forests."

Page 7, line 5-10: "Hence, vegetated coastal habitats would contribute up to 1/3 of the biological $CO_2$ removal by marine biota estimated to represent about 2000 Tg C y-1, which had hitherto been attributed entirely to phytoplankton photosynthesis (Ciais et al 2013). Several points/suggestions regarding this statement:

1/ up to 1/3rd of biological $CO_2$ removal by marine biota: again, this is stressing the upper limit, see comment above. One could take the opposite view and claim they contribute as little as 1/30th ?

Action: As indicated above, I now report the range from minimum to upper estimate (see above).

2/ Unless I'm mistaken, the numbers by Ciais refer to net $CO_2$ uptake by the global ocean, it does not claim that this $CO_2$ drawdown is entirely due to phytoplankton production ?

Comment: The reviewer is incorrect. Although such point is not made explicitly in the text. Examination of the figure (Fig. 6.4) depicting the carbon budget for the global ocean presented by Ciais et al. (2013) shows that the CO2 removal in the ocean is attributed to phytoplankton production. While I do not necessarily agree with that conclusion, this is the depiction that has ben consolidated by the IPCC over several assessments.

Action: I now make this attribution more explicitly by stating: "Hence, vegetated coastal habitats would contribute between a minimum of 0.3% to a maximum of 1/3 of the biological CO2 removal by marine biota estimated to represent about 2,000 Tg C year-1, which had hitherto been attributed entirely to phytoplankton photosynthesis in depictions of the global carbon budget (Fig. 6.4 Ciais et al. 2013).".

2/ Vegetated coastal ecosystems such as mangroves and salt marshes (and subtidal seagrass beds to a certain extent) take up CO2 from the atmosphere, not from the ocean water. Hence, their productivity would not directly lower pCO2 in the ocean and will not lead to CO2 uptake by the ocean, in that context comparing NPP data from all vegetated coastal ecosystems combined is difficult to compare directly with data on net ocean CO2 uptake.

Comment: The reviewer is correct. However, the comparison is not to CO2 uptake by air-sea exchange, but to the attribution in Fig. 6.4 in Ciais et al. (2013) of that uptake to net primary production, as indicated above.

Page 9, line 10: coming back to Smith (1981): while it's good to acknowledge the early work, I doubt this should be used as the most recent / best estimate of the contribution of these ecosystems to NPP. There are many more datasets published in the meantime, and Smith (1981) used a fixed area of 200,000 km2 and included only seagrasses and macroalgae. If the objective is to provide a state-of-the-art, use the best estimates available + include the uncertainty which is a key message elsewhere in the paper.

Action: I now provide the range, from Table 2. Current estimates are still represented

by a range, ecompassing the estimate by Smith 1981), as the uncertainty in area cover persists to date.

-Table2 should be clarified: 1/ "Lower range of production values from Duarte etc": specify whether this refers to the areal rates (first column) or the total NPP range (2nd column).

Action: I agree and have revised the Table heading to indicate: "Net primary production (NPP), carbon burial and export production of vegetated coastal habitats. Lower range of areal production values from Duarte and Chiscano (1999) and upper range of areal seagrass production calculated from gross community production in Gatuso et al. (1998), assuming community respiration (R) R=0.5.GPP from Duarte and Cebrián (1996).".

2/ "Upper value for mangrove and salt marsh production calculated as the ratio between global NPP and global area in Duarte & Cebrian": this does not make sense to me. The global NPP data in Duarte & Cébrian were calculated assuming a certain area for each of these ecosystems, taking those globally integrated NPP values and dividing them by - I assume – a range of (different) area estimates is not defendable. Or perhaps I misunderstand what was done to derive these numbers – explain in more detail.

Action: I have now explained that "Upper value for areal mangrove and salt-marsh production calculated as the ratio between global NPP and global area in Duarte and Cebrián (1996). " Indeed, Duarte and Cebrian (1996) calculated global NPP values by multiplying average areal NPP values – not reported in the paper – by the global area covered, so the calculation reported in the table heading yields the mean areal NPP used by Duarte and Cebrian (1996), and can be recalculated from the published figures.

3/ % buried and exported, data from Duarte & Cébrian (1996). Here too, can these estimates not be easily refined given the large amount of studies performed in the 20 years since this publication ?

Comment: The estimates of % buried and exported, data from Duarte & Cébrian (1996) remain the best estimates reported in the literature. I agree that they could possibly be updated through a new synthesis of published reports, but this would required a dedicated effort, whereas the paper submitted is a review paper. However, I believe the reviewer makes an important point, that many of these estimates have been carried over for much too long and it is time that they be revisited.

Action: I now indicate that incorporating these habitats into depictions of the global ocean budget requires efforts to yield more precise estimates, as those available were produced decades ago. The text now reads: " The carbon fluxes dominated by the "hidden forests" of the coastal ocean are likely to be at least of a similar magnitude and should, therefore, be also captured in future representations of the global carbon budget. This will require an additional effort to improve the precision about current estimates. The uncertainty in the global area these habitats cover has not been narrowed down, for seagrass, macroalgae and salt-marshes, for several decades now, and the estimates of the global NPP contributed by these habitats and their fate have not been revisited since the estimates provided by Smith (1981) and Duarte and Cebrián (1996) several decades ago. As in the case of freshwater carbon emissions and burial, incorporating the carbon fluxes vegetated coastal habitats support into depictions of the global carbon budget and its perturbations also requires that the research community addressing carbon fluxes in vegetated coastal habitats reach out to establish links to share knowledge on these fluxes with the working groups involved in assessing the global carbon budget." ".

Minor issues: -be consistent in using km2 and not Km2

Action: Done.

-Page 2 line6-9: Is an alternative reason not that fitting them into the global C budget is also made complicated by the fact that they are complex ecosystems at the land-ocean interface, and that flux measurements (e.g. OC burial rates in seagrass beds, to name

but one) are somewhat complicated to assign to specific sources/origin, e.g. much of the OC burial in seagrass beds may be terrestrial or mangrove carbon. This paragraph is perhaps a little to pessimistic about the recognition they receive, given the strong impetus in studies on C cycling in vegetated coastal ecosystems during the past 15-20 years.

Action: I agree. The text now reads: "In addition, incorporating marine vegetated coastal habitats into the global carbon budget is made complicated by difficulties in assigning specific sources to the organic carbon burial in their soils, which is often partially allochthonous (e.g. Kennedy et al. 2010)" and "Whereas the focus on Blue Carbon has provided a major impetus to assess the global relevance of marine vegetated coastal habitats in the global carbon budget, these efforts have only addressed the contributions of these habitats to organic carbon burial in sediments, and have addressed other significant contributions of these habitats to the carbon budget of the global ocean.".

-page 8 line 16: poleword (not poelword)

Action: Done.

-page 8 line 9: loss rates of 0.5 – 5 % year-1, this is a different range of loss rates than that cited on page 4 line 9. Use consistent numbers and references.

Action: Done. The text now reads: "The large uncertainties as to the global extent of vegetated coastal habitats are compounded with its rapid change, as these habitats experience some of the steepest rates of any ecosystem, at loss rates of 0.7 to 3 % year-1, depending on ecosystems" throughout.

-page 8, last line: "Lastly, realization of the major export of organic matter [: : :]": what is meant by this?

Action: The sentence has now been clarified. It now reads: "Lastly, evidence of the major export of organic carbon from vegetated coastal habitats to the open ocean

should prompt research, assisted by the available of more and more powerful markers, to elucidate its role in the functioning of the open ocean and deep-sea ecosystems, a role that was considered significant already fifty years ago (cf. Krause-Jensen and Duarte 2016).".

-page 9, first line: "available" should be "availability"

Action: Done.

Changes made in response to comments by Anonymous Referee 2

General comments: This manuscript is a significant addition to the author's previous 4 reviews and syntheses on the role of vegetated shallow coastal ecosystems in carbon sinks (Nellemann et al., 2009; McLeod et al., 2011: Duarte et al., 2013a; Krause-Jensen and Duarte, 2016); in particular, a new inclusion of estimates of "off-site blue carbon sink", i.e., carbon export from coastal habitats and subsequent sequestration in the deep sea (Table 2). The manuscript is generally well-written and easy to follow; however, I found numerous inconsistencies between the citation in the text and the reference list, which should be corrected.

Action: We thank the reviewer for his/her assessment and bringing to my attention the inconsistencies, which I have now corrected.

The result was a surprising because the author estimated that, by including previously overlooked macroalgal contributions, the carbon export far exceeds carbon burial, which is previously thought to be the major mechanism for coastal carbon sequestration. Although, as the author mentioned, this estimate has a large 10-fold uncertainties and not well constrained at this stage, this manuscript would potentially revise our recognition on the role of vegetated coastal habitats on carbon sequestration.

Comment: We thank the reviewer for this assessment of our contribution.

In terms of carbon budget in vegetated coastal habitats, this synthesis still lacks quantification of two boundary conditions (terrestrial inputs and air-water carbon exchanges)
and their effects on the carbon sequestration. However, my understanding is that we have large research gaps in this two components and can be mentioned in the Future Trends and Research Needs section, although several recent papers have been dealing with air-sea CO2 exchanges in such a vegetated shallow waters and other papers have shown the separated estimation of green (terrestrial allochthonous) and blue (autochthonous) carbon in the sediments.

Comment: We agree. We have now expanded the Future Trends and Research Needs sections to address exchanges between the two boundaries (terrestrial inputs and air-water carbon exchanges).

Action: The section Future Trends and Research Needs now includes a paragraph addressing these exchanges, which reads: "Whereas estimates of offshore export of "blue carbon" are now becoming available (Cebrián and Duarte 1986, Duarte et al. 2005, Dittmar et al. 2006, Barrón et al. 2014, 2015, Krause-Jensen and Duarte 2016), the exchange of carbon across the air-sea and land-ocean boundaries of vegetated coastal habitats remains poorly resolved. The strong autotrophic nature of macroalgal and seagrass habitats is further reflected in their role as strong sinks for atmospheric CO2 (e.g. Gazeau et al. 2005, Unsworth et al. 2012, Antony et al. 2013, Tokoro et al. 2014, Ikawa and Oechel 2015). Seagrass, salt-marshes, macroalgae and mangroves all contribute significant loads of material to adjacent beaches, where they can accumulate large carbon stocks (e.g. Mateo et al. 2003, Simeone and de Falco 2012, Gomez et al 2013). However, they receive greater subsidies of plankton and land-derived "green carbon", which have been shown to comprise typically about 50% of the organic carbon stock in seagrass sediments (Kennedy et al. 2010). Hence, organic carbon input from offshore and land sources contribute to the large carbon burial capacity of vegetated coastal habitats while allowing them to export a significant fraction of their own production. Resolving the exchange of carbon between vegetated coastal habitats and adjacent marine, terrestrial and atmospheric components will help further constrain their local and global role in carbon budgets, as well as the consequences of

losses or gains of these habitats on carbon flow".

Specific comments: P17 Table2: I understand that the estimates of Export is new; however, are the estimates of NPP and Burial new? i.e, are these estimates overall the same as the previous your publications (Nellemann et al., 2009; McLeod et al., 2011: Duarte et al., 2013a; Krause-Jensen and Duarte, 2016) and Pendleton et al. (2012) or revised values? If revised, how much is the differences compared to the previous estimates?

Comment: The estimates use a similar basis, but they better acknowledge the uncertainty in area covered used to scale up individual estimates to global figure. Such exercise, of propagating uncertainty in area, had not been done before. The need to account for uncertainty in global area covered is discussed throughout the paper. For instance, the text reads: "Recently, Krause-Jensen and Duarte (2016) propagated uncertainties in the areal extent and primary production of macroalgae to derive an estimate of NPP for macroalgae at 1.52 Pg C year-1, with the 25 and 75 percentiles of this estimate at 1.02 and 1.96 Pg C year-1. However, a similar exercise has not yet been attempted for other vegetated habitat types. Hence, the total net community production of marine vegetated habitats spans a broad 10-fold range from a minimum of 0.4 to 5.4 Pg C year-1 (Table 2), due to combinations of uncertainties in the areal extent, the dominant source of uncertainty, and the average net primary production per unit area. ".

Technical corrections: P2L11, P4L8, P: Duarte et al. 2009 is missing in the Reference

Action: The reference should have been Duarte et al. 2008, this error is now corrected.

P2L26: Duarte et al. 1991 is missing in the Reference

Action: This citation has not been removed.

P2L29: Gattuso et al. 2006, not 2006 but 2005?

Action: Gattuso et al. (2006) now included in references.

P3L19: Chmura et al, 2003 is missing in the Reference

Action: Reference has been added.

P4L10: Waycott et al 2006 is missing in the Reference, 2009?

Action: Reference should be 2009 (error corrected).

P4L12: Whitaker and Likens 1973 is missing in the Reference

Action: Reference has been added.

P5L2: Duarte et al. 2011 is missing in the Reference

Action: Reference has been added (but should be Duarte et al. 2010, corrected).

P5L8: Fourqurean et al. 2014 is missing in the Reference

Action: Reference has been added.

P5L9, P8L10: Duarte et al. 2013, "a" or "b" here?

Action: Now indicated

P5L32: Coupland et al. 2007 is missing in the Reference

Action: Reference has been added.

P7L16: Mazarrasa et al. 2015 is missing in the Reference

Action: Reference has been added.

P7L23: Smith 2013 is missing in the Reference

Action: Reference has been added.

P7L28: May 1994 is missing in the Reference

Action: Mayer 1994 has been added.

P8L12: Waycott et al. 2011 is missing in the Reference, 2009?

Action: Corrected to 2009.

P7L28: Duarte et al. 2002 is missing in the Reference

Action: Reference has been added.

P9L30: Macreadie?

Action: Reference checked.

P10L18: not cited in the text P11L22: not cited in the text P12L4: not cited in the text P13L13: not cited in the text P14L8: not cited in the text P15L4: not cited in the text

Action: References not cited removed from text.